# A Calibrated Simulation for Offline Training of Reinforcement Learning Agents to Optimize Energy and Emission in Office Buildings

## Abstract

Modern commercial Heating, Ventilation, and Air Conditioning (HVAC) systems form a complex and interconnected thermodynamic system with the building and outside weather conditions, and current setpoint control policies are not fully optimized for minimizing energy use and carbon emission. Given a suitable training environment, a Reinforcement Learning (RL) model is able to improve upon these policies, but training such a model, especially in a way that scales to thousands of buildings, presents many practical challenges. To address these challenges, we propose a novel simulation based approach, where a customized simulator is used to train the agent for each building. Our simulator is lightweight and calibrated with recorded data from the building to achieve sufficient fidelity. On a two-story, 68,000 square foot building, with 127 devices, we were able to calibrate our simulator to have just over half a degree of drift from the real world over a 6 hour period. We train an RL agent on this simulator and demonstrate that our agent is able to learn an improved policy. This approach is an important step toward having a real-world Reinforcement Learning control system that can be scaled to many buildings, allowing for greater efficiency and resulting in reduced energy consumption and carbon emissions.

## 1 Introduction

Energy optimization and management in commercial buildings is a very important problem, whose importance is only growing with time. Buildings account for 39% of all US carbon emissions (Lu & Lai, 2020). Reducing those emissions by even a small percentage can have a significant effect. In climates that are either very hot or very cold, energy consumption is much higher, and there is even more room to have a major impact.

Most office buildings are equipped with advanced HVAC devices, like Variable Air Volume (VAV) devices, Hot Water Systems, Air Conditioner and Air Handlers that are configured and tuned by the engineers, manufacturers, installers, and operators to run efficiently with the device's local control loops McQuiston et al. (2023). However, integrating multiple HVAC devices from diverse vendors into a building "system" requires technicians to program fixed operating conditions for these units, which may not be optimal for every building and every potential weather condition. Existing setpoint control policies are not optimal under all conditions, and the possibility exists that a machine learning model may be trained to continuously tune a small number of setpoints to achieve greater energy efficiency and reduced carbon emission.

Our contributions include a highly customizable and scalable HVAC and building simulator, a rapid configuration method to customize the simulator to a particular building, a calibration method to improve this fidelity using real world data, and an evaluation method to measure the simulator fidelity. We also tran an RL agent on the simulator using Soft Actor Critic (Haarnoja et al., 2018), and demonstrate that we can learn an improved policy. This system enables offline learning, where the agent can train in an efficient sandbox environment that adequately emulates the dynamics of the building before being deployed on the real building, an important step towards a real world RL HVAC control system. We first present the simulator, discuss our configuration and calibration tech-

niques, and finally present the results of tuning the simulator and training the RL agent. We also note that our simulator and tuning process will all be made open source.

## 2 OPTIMIZING ENERGY AND EMISSION IN OFFICE BUILDINGS WITH REINFORCEMENT LEARNING

In this section we frame this energy optimization problem in office buildings as a Reinforcement Learning problem (RL). We define the state of the office building St at time t as a fixed length vector of measurements from sensors on the building's devices, such as a specific VAV's zone air temperature, gas meter's flow rate, etc. The action on the building At is a fixed-length vector of device setpoints selected by agent at time t, such as the boiler supply water temperature setpoint, etc. We also define a custom feedback signal, or reward, $R_t(S_t, A_t, v)$ that indicates the quality of taking action $A_t$ in state $S_t$, as a weighted sum of negative cost functions for carbon emission, energy consumption, and zone-level setpoint deviation.

More generally, Reinforcement Learning (RL) is a branch of machine learning that attempts to train an agent to choose the best actions to maximize some long-term, cumulative reward (Sutton & Barto, 2018). As depicted below, the agent observes the state $S_t$ from the environment at time $t$, then chooses action $A_t$. The environment responds by transitioning to the next state $S_t + 1$ and returns a reward (or penalty) after the action, $R_t + 1$. Over time, the agent will explore the action space and learn to maximize the reward over the long term for each given state. A discount factor 1 reduces the value of future rewards amplifying the value of the near-term reward. When this cycle is repeated over multiple episodes, the agent converges on a state-action policy that maximizes the long-term reward.

This sequence is often formalized as the Markov Decision Process (MDP), described by the tuple $(S, A, p, R)$ where the state space is continuous (e.g., temperatures, flow rates, etc.) and the action space is continuous (e.g., setpoint temperatures) and the transition probability $p : S \times S \times A \rightarrow [0, 1]$ represents the probability density of the next state $S_t + 1$ from taking action $A_t$ on the current state $S_t$. The reward function $R : S \times A \rightarrow [R_{min}, R_{max}]$ emits a single scalar value at each time $t$. The agent is acting under a policy $(A_t|S_t)$ which represents the policy of taking action $A_t$ from state $S_t$. The goal of reinforcement learning is to find the policy that maximizes the expected long-term cumulative reward. The set of parameters $\boldsymbol{\theta}^*$ of the optimal policy can be expressed as:

$$\boldsymbol{\theta}^* = \arg\max \theta \mathbb{E}_{\tau \sim \pi\theta(\tau)} \left[ \sum_t \gamma^t R(S_t, A_t) \right]$$

where is the current policy parameter, and is a trajectory of states, actions, and rewards over multiple time steps $t$. In order to converge to the optimal policy, the agent requires many training iterations, making training directly on the building from scratch inefficient, impracticable, if not impossible. Therefore, it is necessary to enable offline learning, where the agent can train in an efficient sandbox environment that adequately emulates the dynamics of the building before being deployed on the actual building.

## 3 RELATED WORKS

Considerable attention has been paid to HVAC control (Fong et al., 2006) in recent years (Kim et al., 2022), and a growing portion of that has considered how Reinforcement Learning and its various associated algorithms can be leveraged (Yu et al., 2021; Mason & Grijalva, 2019; Yu et al., 2020; Gao & Wang, 2023; Wang et al., 2023; Vázquez-Canteli & Nagy, 2019; Zhang et al., 2019b; Fang et al., 2022; Zhang et al., 2019b). As mentioned above, a central requirement in RL is the offline environment that trains the RL agent. Several methods have been proposed, largely falling under three broad categories.

**Data-driven Emulators**: Some works attempt to learn a dynamics as a multivariate regression model from real world data (Zou et al., 2020; Zhang et al., 2019a), often using recurrent neural network architecture, such as Long Short-Term Memory (LSTM) (Velswamy et al., 2017; Sendra-Arranz & Gutiérrez, 2020; Zhuang et al., 2023). The difficulty here is that data-driven models often do not generalize well to circumstances outside the training distribution, especially since they are not physics based.

**Offline RL**: The second approach is to train the agent directly from the historical real world data, without ever producing an interactive environment (Chen et al., 2020; 2023; Blad et al., 2022). While the real world data is obviously of high accuracy and quality, this presents a major challenge, since the agent cannot take actions in the real world and interact with any form of an environment, severely limiting its ability to improve over the baseline policy producing the real world data (Levine et al., 2020).

**Physics-based Simulation**: HVAC system simulation has long been studied (Trčka & Hensen, 2010; Riederer, 2005; Park et al., 1985; Trčka et al., 2009; Husaunndee et al., 1997; Trcka et al., 2007). EnergyPlus (Crawley et al., 2001), a high-fidelity simulator developed by the Department of Energy, is commonly used (Wei et al., 2017; Azuatalam et al., 2020; Zhao et al., 2015; Wani et al., 2019; Basarkar, 2011), but suffers from the scalability issues outlined above.

To overcome the limitations of each of the above three methods, some work has proposed a hybrid approach (Zhao et al., 2021; Balali et al., 2023), and indeed this is the category our work falls under. What is unique about our approach is the use of a physics based simulator that achieves an ideal balance between speed and fidelity, which is sufficient to train an effective control agent off-line.

Various works have also discussed how exactly to apply RL to an HVAC environment, such as what sort of agent to train. Inspired by prior effective use of Soft Actor Critic on related problems (Kathirgamanathan et al., 2021; Coraci et al., 2021; Campos et al., 2022; Biemann et al., 2021), we chose to use a SAC agent.

## 4 SIMULATOR DESIGN CONSIDERATIONS

A fundamental tradeoff when designing a simulator is speed versus fidelity. Fidelity is the simulator's ability to reproduce the building's true dynamics that affect the optimization process. Speed minimizes both simulator configuration time, i.e., the time required to configure a simulator for a target building, and the agent training time, i.e., the time necessary for the agent to optimize its policy using the simulator.

Every building is unique, due to its physical layout, equipment, and location. Fully customizing a high fidelity simulation to a specific target building requires nearly exhaustive knowledge of the building structure, materials, location, etc., some of which are unknowable, especially for legacy office buildings. This requires manual "guestimation", which can erode the accuracy promised by high-fidelity simulation. In general, the configuration time required for high-fidelity simulations limits their utility for deploying RL-based optimization to many buildings. High-fidelity simulations also are affected by computational demand and long execution times.

Alternatively, we developed a fast, low-to-medium-fidelity simulation model that was useful in addressing various design decisions, such as the reward function, and the modeling of different algorithms and for end-to-end testing. The simulation is built on a 2D finite-difference (FD) grid that models thermal diffusion, and a simplified HVAC model that generates or removes heat on special "diffuser" control volumes in the FD grid. For more details on design considerations, see Appendix C.

While the uncalibrated simulator is of low-to-medium fidelity, the key additional factor is data. We collect real world observations from the building we are attempting to simulate, and use that data to finetune the simulator. We believe this approach hits the sweet spot in this tradeoff, allowing us for scalability, while maintaining a high enough level of fidelity to be useful.

## 5 A LIGHTWEIGHT, CALIBRATED SIMULATION

Our goal is to develop a method for applying Reinforcement Learning at scale to commercial buildings. To this end, we put forth the following requirements for this to be feasible: We must have an easily customizable simulated environment to train the agent, with high enough fidelity to be useful. To meet this requirement, we designed a light weight simulator based on finite differences approximation of heat exchange. We proposed a simple automated procedure to go from building floor plans to a custom simulator in a short time, and we designed a fine tuning and evaluation pipeline,

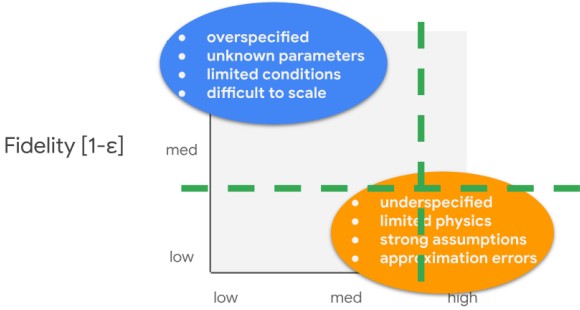

Figure 1: Simulation Fidelity vs. Execution Speed. The ideal operating point for training RL agents for energy and emission efficiency is a tradeoff between fidelity = 1 - normalized error between simulation and real, $\epsilon$ ,and execution speed = number of training steps per second. Additional consideration also includes the time to configure a custom simulator for the target building. While many approaches tend to favor high-fidelity over execution, speed. Our approach argues a low-to-medium fidelity that has a medium-to-high speed is most suitable for training an RL agent.

to use real world data to fine tune the simulation to better match the real world. What follows is a description of our implementation. For a more in depth description, see appendix D.

**Thermal Model for the Simulation** As a template for developing simulators, we propose a general-purpose high-level thermal model for simulating office buildings, illustrated in Figure 1. In this thermal cycle, we highlight significant energy consumers as follows. The boiler burns natural gas to heat the water, $\dot{Q}_b$ . Water pumps consume electricity $\dot{W}_{b,p}$ to circulate heating water through the VAVs. The air handler fans consume electricity $\dot{W}_{b,in}$ , $\dot{W}_{b,out}$ to circulate the air through the VAVs. A motor drives the chiller's compressor to operate a refrigeration cycle, consuming electricity $\dot{W}_c$. In some buildings coolant is circulated through the air handlers with pumps that consume electricity, $\dot{W}_{c,p}$.

We selected **water supply temperature** $\hat{T}_b$ and the **air handler supply temperature** $\hat{T}_s$ as agent actions because they affect the balance of electricity and natural gas consumption, they affect multiple device interactions, and they affect occupant comfort.

**Finite Differences Approximation** The diffusion of thermal energy in time and space of the building can be approximated using the method of Finite Differences (FD)Sparrow (1993); Lomax et al. (2002), and applying an energy balance. This method divides each floor of the building into a grid of three-dimensional control volumes and applies thermal diffusion equations to estimate the temperature of each control volume. By assuming each floor is adiabatically isolated, (i.e., no heat is transferred between floors), we can simplify the three-spatial dimensions into a spatial two-dimensional heat transfer problem. Each control volume is a narrow volume bounded horizontally, parameterized by $\Delta x^2$, and vertically by the height of the floor. The energy balance, shown below, is applied to each discrete control volume in the FD grid, and consists of the following components: (a) the thermal exchange across each face of the four participating faces control volume via conduction or convection $Q_1$, $Q_2$, $Q_3$, $Q_4$, (b) the change in internal energy over time in the control volume $Mc\frac{\Delta T}{\Delta t}$, and (c) an external energy source that enables applying local thermal energy from the HVAC model only for those control volumes that include an airflow diffuser, $Q_{ext}$. The equation is $Q_{ext} + Q_1 + Q_2 + Q_3 + Q_4 = Mc\frac{\Delta T}{\Delta t}$, where $M$ is the mass and $c$ is the heat capacity of the control volume, $\Delta T$ is the temperature change from the prior timestep and $\Delta t$ is the timestep interval. The thermal exchange in (a) is calculated using Fourier's law of steady conduction in the interior control volumes (walls and interior air), parameterized by the conductivity of the volume, and the exchange across the exterior faces of control volumes are calculated using the forced convection equation, parameterized by the convection coefficient, which approximates winds and currents surrounding the building. The change in internal energy (b) is parameterized by the density, and heat capacity of the control volume. Finally, the thermal energy associated with the VAV (c) is equally distributed to all associated control volumes that have a diffuser. Thermal diffusion within the building is mainly ac-

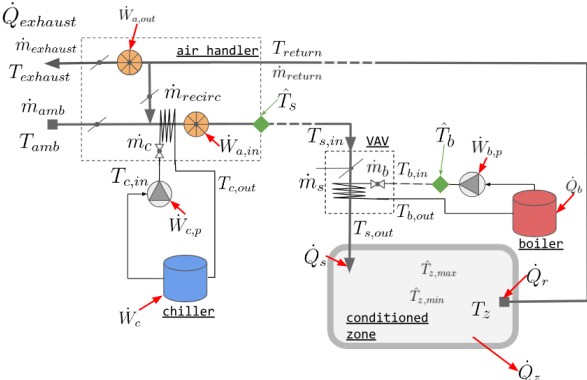

Figure 2: Thermal model for simulation. A building consists of conditioned zones, where the mean temperature of the zone $T_z$ should be within upper and lower setpoints, $\hat{T}_{z,max}$ and $\hat{T}_{z,min}$. Thermal power for heating or cooling the room is supplied to each zone, $\dot{Q}_s$, and recirculated from the zone, $\dot{Q}_r$ from the HVAC system, with additional thermal exchange $\dot{Q}_z$ from walls, doors, etc. The Air Handler supplies the building with air at supply air temperature setpoint $\hat{T}_s$ drawing fresh air, $\dot{m}_{amb}$ at ambient temperatures, $T_{amb}$ and returning exhaust air $\dot{m}_{exhaust}$ at temperature $T_{exhaust}$ to the outside using intake and exhaust fans: $\dot{W}_{a,in}$ and $\dot{W}_{a,out}$. A fraction of the return air can be recirculated, $\dot{m}_{recirc}$. Central air conditioning is achieved with a chiller and pump that joins a refrigeration cycle to the supply air, consuming electrical energy for the AC compressor $\dot{W}_c$ and coolant circulation, $\dot{W}_{c,p}$. The hot water cycle consists of a boiler that maintains the supply water temperature at $T_b$ heated by natural gas power $\dot{Q}_b$, and a pump that circulates hot water through the building, with electrical power $\dot{W}_{b,p}$. Supply air is delivered to the zones through Variable Air Volume (VAV) devices.

complished via forced or natural convection currents, which can be notoriously difficult to estimate accurately. We note that heat transfer using air circulation is effectively the exchange of air mass between control volumes, which we approximate by a randomized shuffling of air within thermal zones, parameterized by a shuffle probability.

**Simulator Configuration** For RL to scale to many buildings, it is critical to be able to easily and rapidly configure the simulator to any arbitrary building. We designed a procedure that enabled a single technician, given floorplans and HVAC layout information, to generate a fully specified simulation in under three hours. In the interest of space, the details are provided in Appendix B.

**Simulator Calibration and Evaluation** In order to calibrate the simulator to the real world using data, we must have a metric with which to evaluate our simulator, and an optimization method to improve our simulator on this metric.

**N-Step Evaluation** We proposed a novel evaluation procedure, based on N-step prediction. Each iteration of our simulator was designed to represent a 5 minute interval, and our real world data is also obtained in five-minute intervals. To evaluate the simulator, we take a chunk of real data of $N$ observations. We then initialize the simulator so that its initial state matches that of the starting observation, and run the simulator for $N$ steps, replaying the same HVAC policy as was used in the real world observations. At this point, we calculate our simulation fidelity metric, which is the mean absolute error of the temperatures in each temperature sensor, at the final timestep. More formally, we define the spatial Mean Absolute Error (MAE) of $Z$ zones at timestep $t$ as:

$$\epsilon_t = \frac{1}{Z} \sum_{z=1}^{Z} |T_{real,t,z} - T_{sim,t,z}| \tag{1}$$

Where $T_{real,t,z}$ is the measured zone air temperature for zone $z$ at timestamp $t$, and $T_{sim,t,z} = \frac{1}{C_z} \sum_{c=1}^{C_z} T_{t,c}$ is the mean temperature of all control volumes $C_z$ in zone $z$ at time $t$.

Table 1: Data used for calibration

| Scenario | Partition | Date | Start Time |
|----------|-----------|------------|------------|
| $S_1$ | train | 2023-07-06 | 16:40 |
| $S_2$ | train | 2023-07-07 | 09:20 |
| $S_3$ | train | 2023-07-08 | 02:00 |
| $S_4$ | train | 2023-07-08 | 18:40 |
| $S_5$ | test | 2023-07-11 | 00:40 |
| $S_6$ | test | 2023-07-11 | 04:40 |
| $S_7$ | test | 2023-07-11 | 09:20 |
| $S_8$ | test | 2023-07-11 | 16:00 |
| $S_9$ | test | 2023-07-12 | 04:40 |

Thus, to evaluate the simulator on $N$-step prediction, we run the simulator for timesteps $0$ to $N-1$, and then calculate the above metric for $t = N - 1$.

**Hyperparameter Calibration**

Once we defined our simulation fidelity metric, the spatial mean absolute temperature error, we can attempt to minimize this error, thus improving fidelity, by hyperparameter tuning several physical constants and other variables.

The parameters varied during the experiment included:

- Forced convection coefficient quantifying outside wind and air currents against the building exterior surfaces.
- Thermal conductivity, heat capacity, and density for exterior and interior walls.
- Shuffle Probability that approximates internal air circulation and interior forced convection.

## 6 EXPERIMENT RESULTS

We now demonstrate the results of how our simulator, when tuned and calibrated, is able to make useful real world predictions.

**Experiment Setup** To test out our simulator, we obtained data on our pilot building, a commercial office building located in northern California. The building has two stories with a combined surface area of 68,000 square feet, and has 127 HVAC devices. We obtained floor plan blueprints and used them to configure a customized simulator for the building, a process that took a single human less than three hours to complete.

**Calibration Data** To calibrate our simulator, we took nine chunks of observations. Four were used to tune the simulator, and the remaining five as validation of the tuned performance on unseen data. All times are given in US Pacific, the local time of the real building. All chunks of time were for 6 hours ($N$-step prediction, where $N = 72$), with the start time listed in table 1.

**Calibration Procedure** We ran hyperparameter tuning for 100 iterations on our simulator. Below are the parameters varied, the ranges given, and the values found that best minimized the calibration metric. The metric was calculated by obtaining the Mean Absolute Spatial Error of temperatures on each of the four train scenarios, and then averaging them.

By including multiple scenarios in our tuning process, we ensured that our hyperparamaters did not overfit to any specific scenario.

Table 2: Hyperparameter ranges and chosen values that best minimized the calibration metric

| Hyperparameter | min | max | best |
|---|---|---|---|
| convection_coefficient ($W/m^2/K$) | 5 | 800 | 255 |
| exterior_cv_conductivity ($W/m/K$) | 0.01 | 1 | 0.93 |
| exterior_cv_density ($kg/m^3$) | 0 | 3000 | 1225 |
| exterior_cv_heat_capacity ($J/Kg/K$) | 100 | 2500 | 100 |
| interior_wall_cv_conductivity ($W/m/K$) | 5 | 800 | 800 |
| interior_wall_cv_density ($kg/m^3$) | 0.5 | 1500 | 0.5 |
| interior_wall_cv_heat_capacity ($J/Kg/K$) | 500 | 1500 | 993 |
| swap_prob | 0 | 1 | 0 |
| swap_radius | 0 | 50 | 50 |

Table 3: Result Metrics

| Metric | Training Data | $S_5$ | $S_6$ | $S_7$ | $S_8$ | $S_9$ | Test Mean |
|---|---|---|---|---|---|---|---|
| MAE | 0.58 $^\circ K$ | 0.45 $^\circ K$ | 1.22 $^\circ K$ | 0.90 $^\circ K$ | 0.65 $^\circ K$ | 0.94 $^\circ K$ | 0.83 |
| Median | NA | -0.29 $^\circ K$ | 1.12 $^\circ K$ | 0.80 $^\circ K$ | -0.49 $^\circ K$ | 0.76 $^\circ K$ | NA |

We reviewed the physical constants that yielded the lowest simulation error from calibration. Densities, heat capacities, and conductivities plausibly matched common interior and exterior building materials. However, the external convection coefficient was higher than under the weather conditions, and likely is compensating for the radiative losses and gains, which were not directly simulated.

**Calibration Results**

We present the predictive results of our calibrated simulator, on $N$-step prediction, where $N = 72$, representing a 6 hour predictive window. We calculated the spatial mean absolute temperature error, as defined above. We also present a second metric, the median spatial temperature error. This was not used in the tuning process, but gives us some insight into how well the calibration process is performing.

As indicated in Table 3, our tuning procedure drifts only $0.58$ degrees on average over a six hour period on the tuning set, and we get good generalization, average test error is only slightly larger at $0.83$. One interesting point is that the performance on scenario $S_6$ and $S_9$ were so different, despite being the same time of day. We believe this is due to the differing weather conditions on those two days affecting our model differently. It should be noted that an uncalibrated model (ie our baseline) had a much larger mean error of $1.97$ degrees.

**Visualizing Temperature Drift Over Time** Figure 3 illustrates temperature drift over time for scenario $S_6$. At each time step, we calculate the spatial temperature error for all sensors, and present the errors as a boxplot distribution.

In this case, this figure is very useful in helping us understand why the error in scenario $S_6$ was larger than the others. The scenario began at 4:40 AM, and at 7:AM, two hours and twenty minutes in, the building shifted from night mode to day mode. As can be observed in the figure, at the two hour twenty minute mark, the simulator began drifting from the real world, becoming a biased estimator.

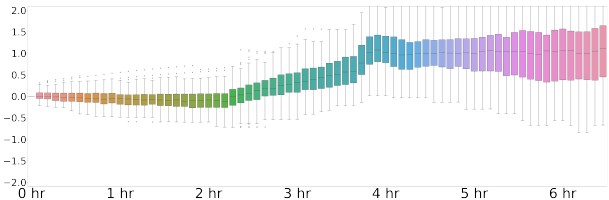

Figure 3: Temperature Drift Over 6 hours, scenario $S_6$. At each time step, the spatial temperature error distribution is shown as a boxplot.

The larger error here was caused by our simulator being unable to correctly predict the transition from night mode to day mode. For similar visuals on the other scenarios, see appendix A.

**Visualizing Spatial Errors** Figure 4 illustrates the results of this predictive process over a six hour period, on scenario $S_4$.

Figure 4.a displays a heatmap of the spatial temperature difference throughout the building, between the real world and simulator, after three hours. Red indicates that in that location, the simulator was warmer than the real world, and blue indicates colder. White means that the simulator and real world were the same temperature in that location. Figure 4.b illustrates the same after six hours. There are several key takeaways from this illustration. The ring of blue around the building indicates that our simulator is too cold on the perimeter, which implies that the heat exchange with the outside is happening more rapidly than it would in the real world. The inside of the building remains red, which means that despite the simulator perimeter being cooler than the real world, the inside is warmer. This implies that our thermal exchange within the building is not as rapid as that of the real world. We suspect that this may be because our simulator does not have a radiative heat transfer model. Lastly, there is a band of white around the perimeter of the building, inside the band of blue and surrounding the red areas. These are the locations where the temperature reached the correct equilibrium, despite too much exterior exchange, and not enough air moving around inside. These locations were the right distance within the building to be the exact same as in the real world data.

As mentioned above, we suspect the culprit is a missing radiative heat model that drives the interior exchange to be too low. To cope with this, the hyper parameter tuning process chooses a too high exterior exchange. These insights can help improve the fidelity of the simulator in the future. For similar visuals of the test scenarios, see appendix A. For a more elaborate version of this visual also including the differences after three hours for comparison, see appendix E.

**Training a Reinforcement Learning Agent** To demonstrate the usefulness of our simulator, we trained a Soft Actor Critic (SAC) agent (Haarnoja et al., 2018) on our simulator, recording the actor loss, critic loss, and return over time [1].

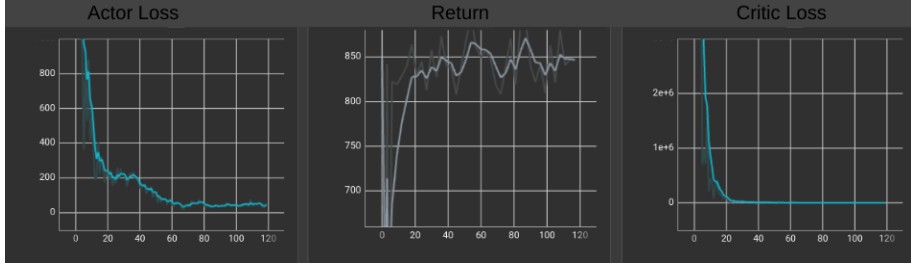

Figure 5: SAC agent metrics

---

[1]This agent was not trained on a simulator using the hyperparameters obtained in the tuning process, due to time constraints, but instead was trained on our simulator with default hyperparameters. We intend to replace this graphic with an equivalent one using the exact same hyperparameters as obtained in tuning for the published draft.

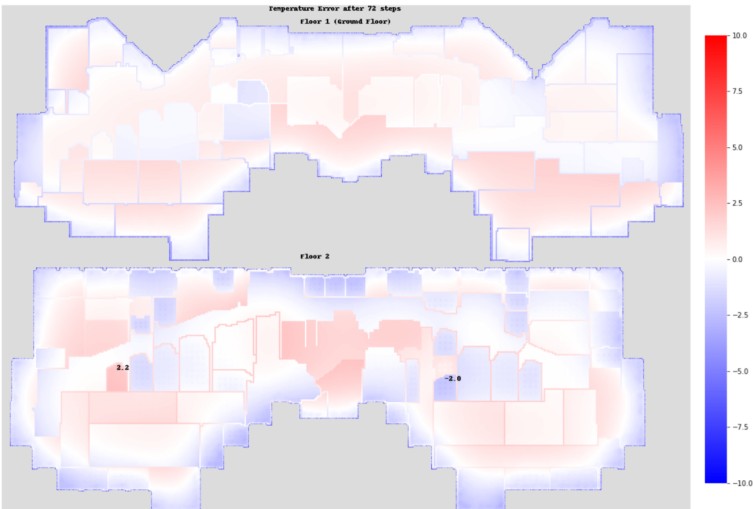

Figure 4: Visualization of simulator drift after 6 hours, on scenario $S_4$. The image is a heatmap representing the temperature difference between the simulator and the real world, with red indicating the simulator is hotter, and blue indicating it is colder. The zone with the max and min temperature difference are indicated by displaying above them the difference.

As can be seen, losses are decreasing and return increasing, indicating the agent is learning some sort of meaningful policy. The replay buffer was first populated with a random policy, and evaluated for 1000000 iterations. Both the actor and critic were feedforward networks, with an actor learning rate of 0.0004, and a critic learning rate of 0.0008.

## 7 DISCUSSION AND FUTURE WORK

While developing a realistic simulator and training an agent is an important step, our long term goal is to successfully deploy an agent to the real world. Thus, our aim for the future is real world transfer, demonstrating that an agent trained using our simulation procedure can indeed transfer to a real building. We already have a real building that we have the ability to control, and using our pipeline produced a simulator that perfectly matches all of its devices, setpoints and sensors. We also have a method of tuning this simulator, based on real data, to make higher accuracy predictions. Our future goal, showing how an agent, trained on this simulator, is able to produce a useful policy on the real building, will require a few more components. In future live trials on the real building, we will train an RL agent on our simulator, and carefully measure its performance in the real world. We also have outlined several approaches that we intend to use to further improve our simulator and fine tuning process

1. Tuning on data from multiple seasons, various times of day, and longer time windows, to improve generalizability and reduce overfitting.

2. Updating our simulation by adding in a radiative heat transfer model to help with interior heat exchange, and by increasing the action space to include air handler and water system device on/off commands and air handler static pressure and hot water system differential pressure.

3. Adding a Neural Network based predictor corrector to the tuning process to further improve fidelity.

In addition, in the future we hope to incorporate an occupancy model (Peng et al., 2018) to give the agent specific knowledge of where the building occupants are located.

We are optimistic that our novel simulation tuning process will allow us to develop a useful, and most importantly, transferable, HVAC control solution, and we hope our work will inspire further effort in this field.

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

## A    EXTRA VISUALS

We include below figures representing the temperature drift over time, and temperature difference after 6 hours, for each of the test scenarios:

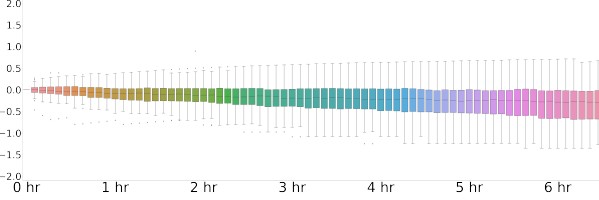

Figure 6: Temperature Drift scenario $S_5$

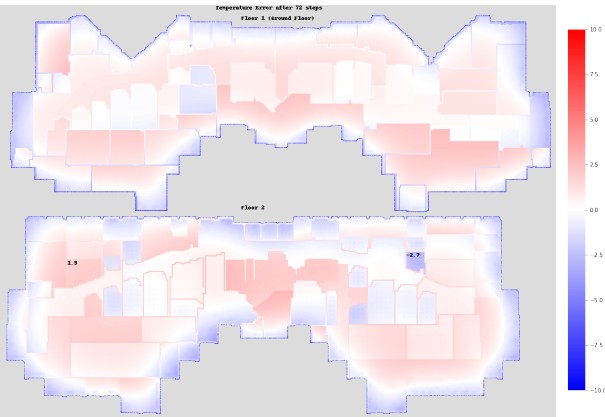

Figure 7: Temperature Difference scenario $S_5$

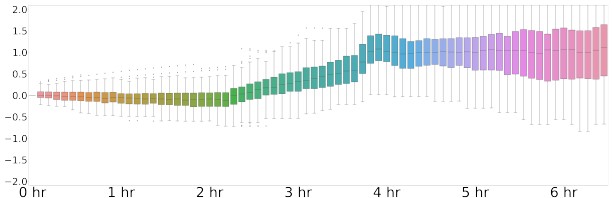

Figure 8: Temperature Drift scenario $S_6$

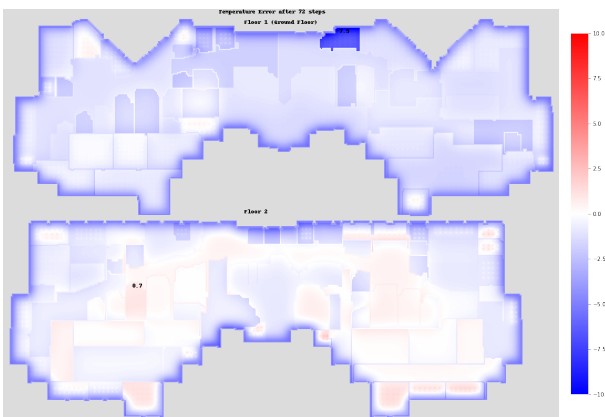

Figure 9: Temperature Difference scenario $S_6$

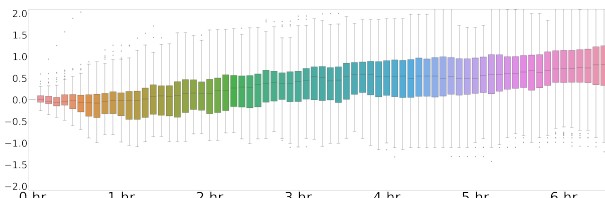

Figure 10: Temperature Drift scenario $S_7$

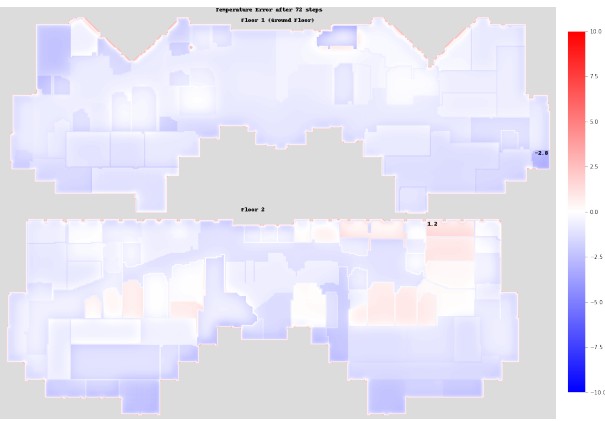

Figure 11: Temperature Difference scenario $S_7$

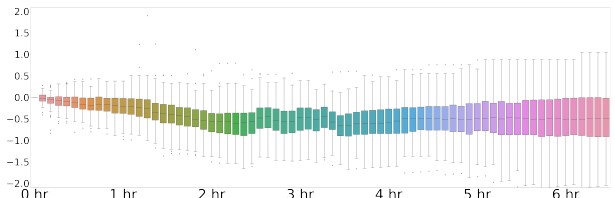

Figure 12: Temperature Drift scenario $S_8$

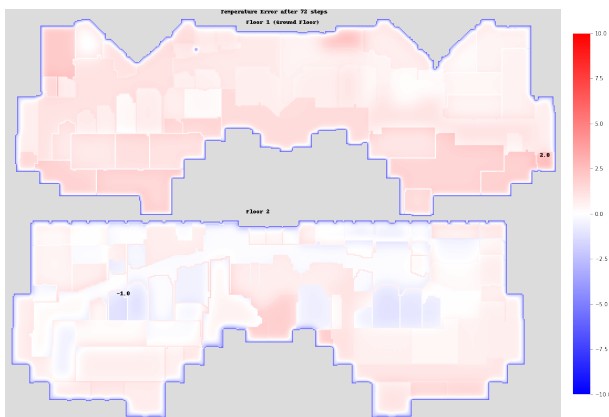

Figure 13: Temperature Difference scenario $S_8$

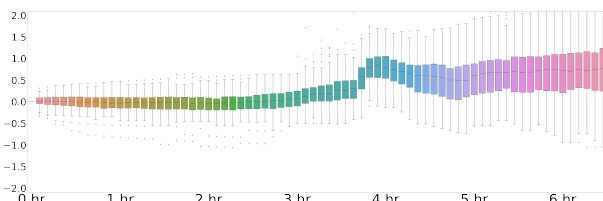

Figure 14: Temperature Drift scenario $S_9$

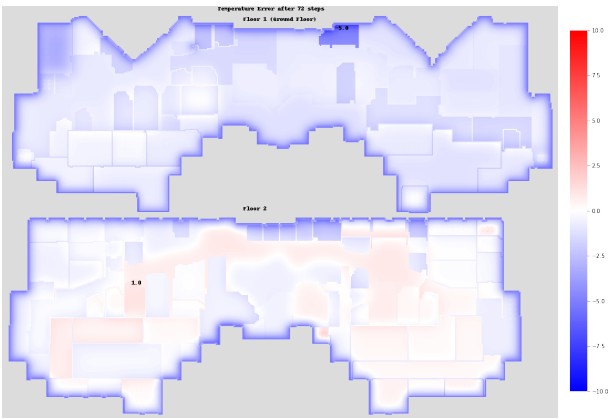

Figure 15: Temperature Difference scenario $S_9$

## B    SIMULATOR CONFIGURATION PROCEDURE DETAILS

To configure the simulator, we require two type of information on the building:

1. Floorplan blueprints. This includes the size and shapes of rooms and walls for each floor.

2. HVAC metadata. This includes each device, its name, location, setpoints, fixed parameters and purpose.

We preprocess the detailed floorplan blueprints of the building, and extract a grid that gives us an approximate placement of walls and how rooms are divided. This is done via the following procedure:

1. Using threshold $t$, binarize the floorplan image into a grid of 0s and 1s.

2. Find and replace any large features that need to be removed (such as doors, a compass, etc)

3. Iteratively apply standard binary morphology operations (erosion and dilation) to the image to remove noise from background, while preserving the walls.

4. Resize the image, such that each pixel represents exactly one control volume

5. Run a connected components search to determine which control volumes are exterior to the building, and mark them accordingly

6. Run a DFS over the grid, and reduce every wall we encounter to be only a single control volume thick in the case of interior wall, and double for exterior wall

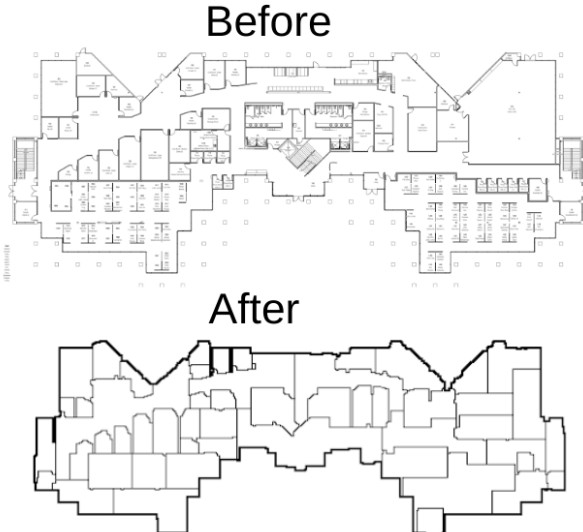

Figure 16: Before and after images of the floorplan preprocessing algorithm

We also employ a simple user interface to label the location of each HVAC device on the floorplan grid. This information is passed into our simulator, and a custom simulator for the new building, with roughly accurate HVAC and floor layout information, is created. This allows us to then calibrate this simulator using the real world data, which will now match the simulator in terms of device names and locations.

We tested this pipeline on our pilot building, which consisted of two floors with combined surface area of 68,000 square feet, and has 127 HVAC devices. Given floorplans and HVAC layout information, a single technician was able to generate a fully specified simulation in under three hours. This customized simulator matched the real building in every device, room, and structure.

## C    SIMULATOR DESIGN CONSIDERATION DETAILS

A simulator models the physical system dynamics of the building, devices, and external weather conditions, and can train the control agent interactively, if the following desiderata are achieved:

1. The simulation must produce the same observation dimensionality as the actual real building. In other words, each device-measurement present in the real building must also be present in the simulation.

2. The simulation must accept the same actions (device-setpoints) as the real building.

3. The simulation must return the reward input data described above (zone air temperatures, energy use, and carbon emission).

4. The simulation must propagate, estimate, and compute the thermal dynamics of the actual real building and generate a state update at each timestep.

5. The simulation must model the dynamics of the HVAC system in the building, including thermostat response, setpoints, boiler, air conditioning, water circulation, and air circulation. This includes altering the HVAC model in response to a setpoint change in an action request.

6. The time required to recalculate a timestep must be short enough to train a viable agent in a reasonable amount of time. For example, if a new agent should be trained in under three days (259,200 seconds), requiring 500,000 steps, the average time required to update the building should be 0.5 seconds or less.

7. The simulator must be configurable to a target building with minimal manual effort.

    We believe our simulation system meets all of these listed requirements.

## D    SIMULATION IMPLEMENTATION DETAILS

### D.1    THERMAL MODEL FOR THE SIMULATION

As a template for a developing a large number of simulators, we propose a general-purpose high-level thermal model for simulating office buildings, illustrated in Figure 2. The primary objective of the climate control system is to maintain the room temperature $T_z$ of each conditioned zone with in its heating and cooling setpoints[2], $\hat{T}_{z,min}$ and $\hat{T}_{z,max}$. The Variable Air Volume (VAV) devices feed the zones with continuous thermal energy[3], $\dot{Q}_s$ and offset thermal losses or gains from windows, floors, people, computers and other heat-generating devices, etc. $\dot{Q}_z$ and the rate of thermal energy drawn out of the zone by recirculation $\dot{Q}_r$. The VAVs regulate the supply airflow, $\dot{m}_s$, and the heating water flow, $\dot{m}_b$ that heats the supply air to a desired temperature $T_{s,out}$ . Boilers supply VAVs with heating water, circulated by electrical water pumps. The air handlers supply multiple VAVs/zones with fresh and potentially cooled air, and regulate the proportion of recirculated and fresh air by setting dampers and running supply and exhaust electrical fans. When cooling is necessary, the air handler cools the supply air with electrically driven chillers and possible coolant pumps. In this thermal cycle, the following significant energy consumers are highlighted:

- The boiler burns natural gas to heat the water, $\dot{Q}_b$ .

- Water pumps consume electricity $\dot{W}_{b,p}$ to circulate heating water through the VAVs.

- The air handler fans consume electricity $\dot{W}_{b,in}$ , $\dot{W}_{b,out}$ to circulate the air through the VAVs.

- A motor drives the chiller's compressor to operate a refrigeration cycle, consuming electricity $\dot{W}_c$ .

- In some buildings coolant is circulated through the air handlers with pumps that consume electricity, $\dot{W}_{c,p}$.

---

[2]We use hat notation to denote action variables.

[3]We use a dot to denote the time derivative, such as energy applied over time (i.e. power).

All electrical consumers may be supplied by electricity obtained from renewable sources. However, the boiler consumes natural gas and contributes to greenhouse gas emissions. The cost of electricity varies by time and current load on the grid; prices are highest during peak demand periods.

### D.2 OPTIMIZATION TRADEOFFS

In a **heating condition** ($T_{amb} < T_{z,min}$), heated air must be fed to the zones to compensate for thermal losses, such that $T_{s,out} > T_z$, and the amount of heat applied is proportional to the mass flow rate times the difference in temperature, $\dot{Q}_s \propto \dot{m}_s \times (T_{s,out} - T_z)$. The agent can trade off air flow for temperature to get the desired heating. The air flow is governed by the fan use and damper configuration, and the supply air temperature is governed by the heating water temperature and the amount of heat applied to the boiler. In a cooling condition ($T_{amb} > T_{z,min}$), heat must be removed from the zone by replacing higher temperature air with lower temperature air, $T_{s,out} < T_z$. In a single-zone configuration, the problem is fairly simple: disable the boiler and engage the chiller until the zone temperature is within the setpoint. However, an air handler must service multiple zones simultaneously with different thermal gains. Some zones that have lower thermal gains, i.e. are colder and require less cooling, may require supplemental heating to ensure that all zones are within the setpoint ranges. Here, the tradeoff is how much cold air needs to be circulated to ensure all zones are within set point while minimizing the amount of heating.

### D.3 SETPOINTS AND ACTIONS

On its initialization, the environment queries the building for device setpoints. Every continuous-valued device setpoint is eligible to be acted upon by an agent action. However, it may not be desirable for the agent to control every possible setpoint reported by the building. Larger action spaces generally are more difficult to optimize than smaller action spaces, and because exploration of some setpoints may jeopardize occupancy comfort (e.g., zone air temperature setpoints), not all setpoints should be controlled by the agent (at least for the initial versions). Furthermore, some device setpoints are controlled by tuned equipment-level controls, and attempting to control these setpoints may affect the device's ability to function properly.

We selected setpoints for actions based on the following criteria:

1. The setpoints should affect the balance of electricity and natural gas consumption.

2. The setpoints should affect device interactions (i.e., changing the setpoint on a device will force a change on another device).

3. Changes to the setpoint should have an indirect effect on occupant comfort.

Based on this criteria, we selected the **water supply temperature** and the **air handler supply temperature** as initial setpoints for agent actions. Since the target building has two primary air handler units and one hot water system, the action space dimensionality is 3.

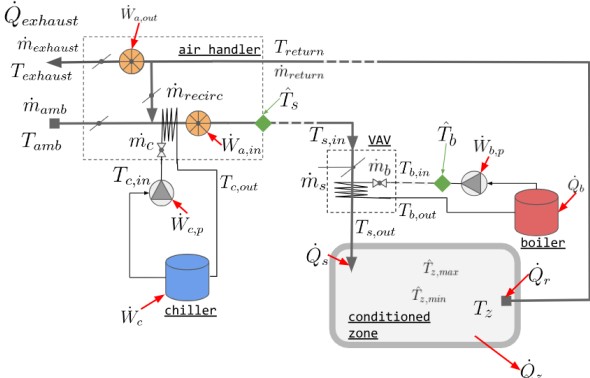

Figure 17: Thermal model for simulation. A building consists of one or more conditioned zones, where the mean temperature of the zone $T_z$ should be within upper and lower setpoints, $\hat{T}_{z,max}$ and $\hat{T}_{z,min}$. Thermal power for heating or cooling the room is supplied to each zone, $\dot{Q}_s$, and recirculated from the zone, $\dot{Q}_r$ from the HVAC system, with additional thermal exchange $\dot{Q}_z$ from walls, doors, windows, etc. The Air Handler supplies the building with air at supply air temperature setpoint $\hat{T}_s$ drawing fresh air, $\dot{m}_{amb}$ at ambient temperatures, $T_{amb}$ and returning exhaust air $\dot{m}_{exhaust}$ at temperature $T_{exhaust}$ to the outside using intake and exhaust fans that consume electrical energy: $\dot{W}_{a,in}$ and $\dot{W}_{a,out}$. A fraction of the return air can be recirculated, $\dot{m}_{recirc}$ for efficiency. Central air conditioning is achieved with a chiller and pump that joins a refrigeration cycle to the supply air, consuming electrical energy for the AC compressor $\dot{W}_c$ and coolant circulation, $\dot{W}_{c,p}$. The hot water cycle consists of a boiler that maintains the supply water temperature at $T_b$ heated by natural gas power $\dot{Q}_b$, and a pump that circulates hot water through the building, with electrical power $\dot{W}_{b,p}$. Supply air is delivered to the zones through Variable Air Volume (VAV) devices that provide zone-level heating with a water-to-air heat exchanger.

### D.4  FINITE DIFFERENCES APPROXIMATION

The diffusion of thermal energy in time and space of the building can be approximated using the method of Finite Differences (FD)(Sparrow, 1993; Lomax et al., 2002), and applying an energy balance. This method divides each floor of the building into a grid of three-dimensional control volumes and applies thermal diffusion equations to estimate the temperature of each control volume. By assuming that each floor is adiabatically isolated, (i.e., no heat is transferred down or up between floors), we can simplify the three-spatial dimensions into a spatial two-dimensional heat transfer problem. Each control volume is a narrow volume bounded horizontally, parameterized by $\Delta x^2$, and vertically by the height of the floor.

The energy balance, shown in Equation 1, is applied to each discrete control volume in the FD grid consists of the following components: (a) thermal exchange across each face of the four participating faces control volume via conduction or convection $Q_1, Q_2, Q_3, Q_4$, (b) the change in internal energy over time in the control volume $Mc\frac{\Delta T}{\Delta t}$, and (c) an external energy source that enables applying local thermal energy from the HVAC model only for those control volumes that include an airflow diffuser, $Q_{ext}$.

$$Q_{ext} + Q_1 + Q_2 + Q_3 + Q_4 = Mc\frac{\Delta T}{\Delta t} \tag{2}$$

Where $M$ is the mass and $c$ is the heat capacity of the control volume, $\Delta T$ is the change in temperature from the previous time step and $\Delta t$ is the timestep interval.

The thermal exchange in (a) is calculated using Fourier's law of steady conduction in the interior control volumes (walls and interior air), parameterized by the conductivity of the volume, and the exchange across the exterior faces of control volumes are calculated using the forced convection equation, parameterized by the convection coefficient, which approximates winds and currents sur-

rounding the building. The change in internal energy (b) is parameterized by the density, and heat capacity of the control volume. Finally, the thermal energy associated with the VAV (c) is equally distributed to all associated control volumes that have a diffuser.

Thermal diffusion within the building is mainly accomplished via forced or natural convection currents, which can be notoriously difficult to estimate accurately. We note that heat transfer using air circulation is effectively the exchange of air mass between control volumes, which we approximate by a randomized shuffling of air control volumes within thermal zones, parameterized by a shuffle probability.

# E   THREE HOUR SIX HOUR TEMPERATURE DIFFERENCE COMPARISON

The figure below shows a comparison of temperature difference, on scenario $S_4$.

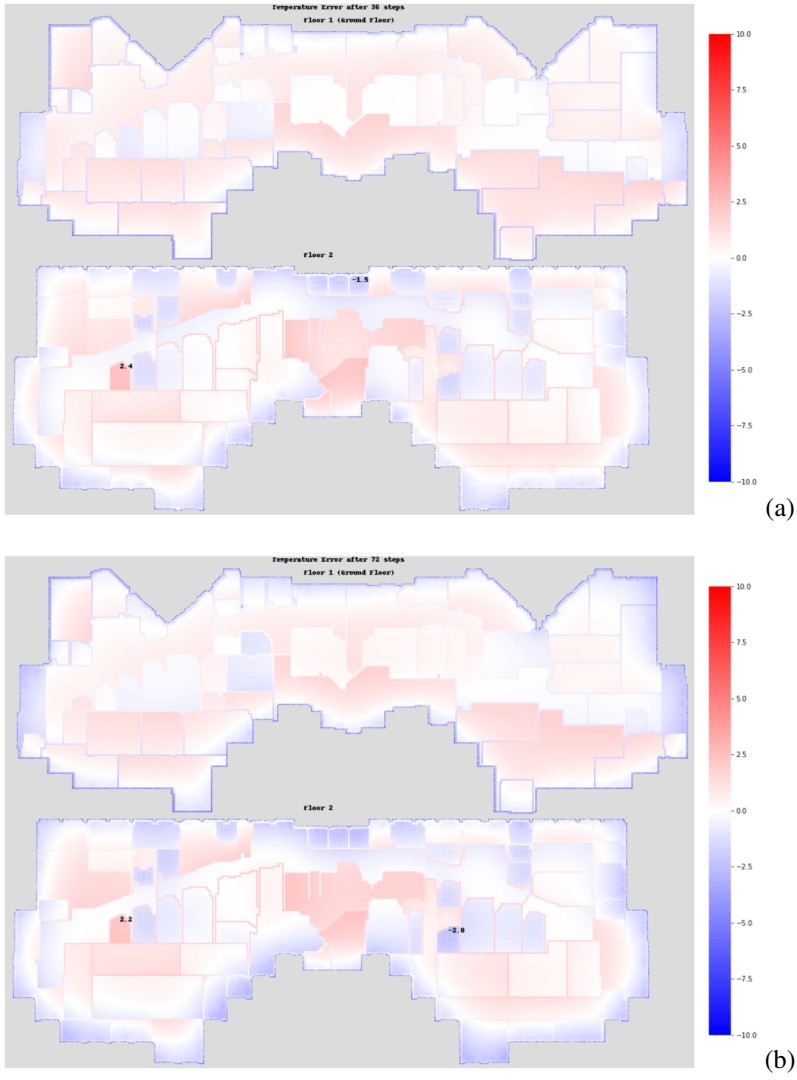

Figure 18: Visualization of simulator drift after 3 and 6 hours, on scenario $S_4$. The image is a heatmap representing the temperature difference between the simulator and the real world, with red indicating the simulator is hotter, and blue indicating it is colder. The zone with the max and min temperature difference are indicated by displaying above them the difference.

The ring of blue around the building indicates that our simulator is too cold on the perimeter, which implies that the heat exchange with the outside is happening more rapidly than it would in the real world, and we can see this difference is more exaggerated after six hours than after three.

Our thermal exchange within the building is not as rapid as that of the real world. We suspect that this may be because our simulator does not have a radiative heat transfer model. We can also observe that this is actually better after six hours than three, since the interior air has had more time to exchange.

