# OpenReview forum: "A Calibrated Simulation for Offline Training of Reinforcement Learning Agents to Optimize Energy and Emission in Office Buildings"
_ICLR.cc/2024/Conference — Submitted to ICLR 2024_

### Official Review · Reviewer_BrjB · 2023-10-14

**Soundness:** 3 good
**Presentation:** 1 poor
**Contribution:** 2 fair
**Rating:** 3
**Confidence:** 4

**Summary:**

This paper propose a new simulation based approach, combined with RL, to solve the problem of energy minimization of Heating, Ventilation, and Air Conditioning (HVAC) systems. An RL agent is trained and demonstrate a well-worked policy.

**Strengths:**

- The problem studied is meaningful. And the design of the simulator seems novel and well worked.

- Experimental results seems good, but lack of comparison, making it hard to know if this is the best solution to this problem.

**Weaknesses:**

- No baselines included, hard to know the contribution and advantage of this work.

- Some related work missed. [1][2]

- The writing of this paper is bad, making it hard to make sure what this work contributes. And there are too many typos, please carefully revise them, and also the layout problem.
    - Found typos / layout problems:
    - Sec1, para3: We also **tran** an RL agent on the simulator.
    - Sec2, next state $s_t+1$ $R_t+1$ should be $s_{t+1}$ $R_{t+1}$.
    - Use math type for every timestep $t$ please.
    - Sec5, Figure 1 should be Figure 2. The authors should learn to use \ref in Latex.
    - All figures should be in a format of pdf to make it clear. The font in Figure 4 is too small (similar in Appendix) and Figure 5 should be plot with better tools instead of tensorboard screen shot, which is not clear about what the x-y axis means.

- See questions for more.

[1] Zhang W, Yang Z, Shen J, et al. Learning to build high-fidelity and robust environment models[C]//Machine Learning and Knowledge Discovery in Databases. Research Track: European Conference, ECML PKDD 2021, Bilbao, Spain, September 13–17, 2021, Proceedings, Part I 21. Springer International Publishing, 2021: 104-121.
[2] Xu T, Li Z, Yu Y. Error bounds of imitating policies and environments[J]. Advances in Neural Information Processing Systems, 2020, 33: 15737-15749.

**Questions:**

1. Is "A discount factor 1 reduces the value of future rewards amplifying the value of the near-term reward." a typo where $1$ should be $gamma$? This make me confused since a discounted factor 1 will amplify the future reward instead of reducing the value of future rewards, and normally RL algorithms requires to be converged with a discounted factor $0 < \gamma < 1$

2. I am not familiar with the problem of energy optimization in office buildings, what is the action exactly? I mean, given it is a vector of device setpoints, how it performs on the building energy?

3. Is the thermal model mentioned in Sec 5 a novel proposal or is referred to some related work? I did not see any reference here.

4. Sec 2 says the agent action is `a vector of device setpoints`, and Sec 5 says `water supply temperature and the air handler supply temperature s as agent actions`. What is the action exactly? Can you be more specific and connect these context?

5. The paper argue that their simulator is `lightweight and calibrated`, and `balance between speed and fidelity`, yet I did not get why this work make a better balance, can you make this statement more clear and highlight what is the difference of this work compared with previous ones in detail?

6. How do you do `Hyperparameter Calibration`? If you just tune it by hand, then what is the contribution?

7. In abstract, `learn an improved policy`, improved upon what?

I may change my evaluation when the paper solves my concern.

---

### Official Review · Reviewer_doaN · 2023-10-27

**Soundness:** 2 fair
**Presentation:** 1 poor
**Contribution:** 1 poor
**Rating:** 1
**Confidence:** 5

**Summary:**

This paper presents a novel simulator-based calibration approach for offline training of reinforcement learning agents to optimize energy and emission in office buildings. Specifically, the authors use 2D finite difference method to represent the building HVAC simulator and then utilize data collected from a real building to train a RL agent in an offline manner on top of such a simulation environment. To validate the propose method, the authors collect data from a pilot building and show an improved policy.

**Strengths:**

The investigated topic is interesting, particularly deploying RL agent to real buildings and scale it in an efficient way remaining a very challenging task. The proposed approach enables a lighweight simulator only calibrated with a reasonably amount of data.

**Weaknesses:**

This paper still requires a substantial amount of work to improve its quality. In particular, the authors need to address the following issues.

1. The lack of quantitative technical detail and contents. Overall, the paper was written in a qualitative way such that it requires many technical clarifications in RL settings, specific reward function, HVAC simulation model, notations, etc.

2. No justification for simulation model simplification. Though the authors claim their simulator is lightweight, they need simplification on their simulation model. For example, their simulation is built on 2D finite-difference grid. Such a grid-based simulation can bring approximation error in the RL. How did the authors avoid that? Additionally, there is no heat transferred between floors. Is this assumption realistic? There should be heat transferred between floors typically. How to justify these conditions or assumptions in the simulation model?

3. No baseline comparison in the experimental results. The authors did really mention some existing methods. But there is no baseline comparison in the work such that we have no idea whether the proposed method has superiority over them.

4. Only one pilot building is not sufficient to validate the scheme. Also, as indicated in the title, the authors mentioned energy and emission. But the results are based on temperature, what is the connection? I didn't see the optimization in terms of energy and emissions.

5. The main contents should be self-explained. The authors should present sufficient technical detail to make sure it is clear to readers.

6. The authors need typo check to improve the writing.

Overall, this paper is not technically solid and sound.

**Questions:**

Please see the questions in the above weaknesses section.

---

### Official Review · Reviewer_P7Wf · 2023-11-02

**Soundness:** 2 fair
**Presentation:** 1 poor
**Contribution:** 1 poor
**Rating:** 1
**Confidence:** 5

**Summary:**

Paper proposes a simulation system for building HVAC systems which can be used to train reinforcement learning based control agents. The simulation uses physics-based approach for simulation, overcoming the limitations of ML based system that do not generalize to out-of-distribution data. Traditional physics-based simulations are too slow and computationally expensive, which make RL agents difficult to train. The simulator is calibrated to real-world data using hyper-parameter tuning of physics equations. An RL agent is trained to show the feasibility of the simulation system.

**Strengths:**

- The problem statement is clear, and is a well-known problem in this domain
- The idea makes sense, using fast physics-based simulations overcomes the limitations of prior simulation systems like EnergyPlus
- A simulator of a real world building that can be created in just 3 hours is appealing

**Weaknesses:**

- Paper claims that the simulations will generalize to out-of-distribution data as it is based on physics, however no evidence has been provided to show the generalization. As a result, it is unclear if the RL agent is going to learn a meaningful policy.
- The drift in error accumulates over time, and has been shown in the evaluation period of 6 hours. RL agents require simulation for at least a year, it is unclear how the simulation system can be practically useful.
- No details of the RL agent simulation is provided. What is the episode length? What were the weather conditions? What were the occupancy conditions?
- The main contribution of the paper is based on physics equations of heat transfer. However, they have not been explained adequately. Precise description of the equations used is required to assess the fidelity of the physics simulations.
- Figure 3 indicates that the control system in the simulation has not been properly modeled. The control system should capture the transition from day to night mode.
- It is unclear if an error of 0.83 degrees is meaningful. First, it's unclear if the unit is Celsius or Fahrenheit. Second, it is unclear how much the indoor temperature drifts during a 6 hour period. If the range of temperature is only 2 degrees, then an error of 0.83 is >40%.
- Unclear why certain aspects like radiative gains were ignored in the simulator.

**Questions:**

Weaknesses above summarize the questions I have for the authors. Brief summary below
- What are the details of the RL simulation system? How do you know if the simulation is accurate? I would have expected a controlled experiment where the HVAC settings are changed and the corresponding measurements are matched against the simulations.
- What are the equations used in the simulator?
- What is the range of temperature values in the train and test dataset?
- How long do you run the simulation while training the RL agent? Is the error accumulation sufficiently low to train an agent during that period?

---

### Official Review · Reviewer_ffyP · 2023-11-10

**Soundness:** 3 good
**Presentation:** 2 fair
**Contribution:** 2 fair
**Rating:** 5
**Confidence:** 4

**Summary:**

This paper provides an RL simulator for HVAC control in buildings, which can be better tuned for a specific building using real data about the building. The paper describes the simulator, briefly describes the process for calibrating and evaluating the simulator based on real-world data, assesses the fidelity of the resultant simulator compared to measured outcomes on a real building, and demonstrates the training of an RL agent on this simulator.

**Strengths:**

The area of work addressed by the paper is very much needed - HVAC is one of the largest contributors of emissions in buildings, but simulators that are meant to incentivize the development of RL for efficient HVAC control do not generalize between buildings.

The fidelity of the proposed (post-tuning) simulator for the building evaluated is impressive.

**Weaknesses:**

In its current form, the work is too early for submission to ICLR. In particular, while the main contribution posed is generalization across buildings, only one building is evaluated in this work. Evaluation of two or more buildings would be necessary to understand whether the simulator tuning procedure actually works well across buildings, or whether some of the simulator's assumptions (e.g., its underlying physical specifications) prevent generalization.

In addition, it would be important to understand whether the RL agent trained on this simulator actually ends up working well on the real building (which the authors flag as future work, but I think is necessary to assess the strength of the present contribution).

Is it possible to also tune EnergyPlus using the same simulator tuning procedure? If so, a comparison to that option should be included.

There is not enough discussion in the main body of the paper about the structure of the simulator or the tuning procedure. Given that the main contribution posed (beyond the simulator itself) is the tuning procedure, more details should be provided.

Minor: There are many typos in both the prose and math, which should be fixed.

**Questions:**

* Does the simulator tuning procedure actually generalize across buildings?
* Does the RL agent trained on the simulator work well on the real building?
* How well does it work to tune EnergyPlus based on real building data, as an alternative to training the simulator proposed by the authors? Has this been done before?

---

### Meta-Review · Area_Chair_3ebS · 2023-12-12

**Metareview:**

This paper studies HVAC control in office buildings and provides a new RL simulator. Although the reviewers acknowledge the contributions of this work, the lack of baselines and necessary experiments as well as the missing details make this work below the threshold of acceptance. The authors also fail to rebuttal. Therefore, I recommend rejection.

**Justification For Why Not Higher Score:**

The lack of baselines and necessary experiments, some important details are missing.

**Justification For Why Not Lower Score:**

N/A

---

### Decision · Program_Chairs · 2024-01-16

Reject